# A Boolean Task Algebra For Reinforcement Learning

**Geraud Nangue Tasse, Steven James, Benjamin Rosman**
School of Computer Science and Applied Mathematics
University of the Witwatersrand
Johannesburg, South Africa
geraudnt@gmail.com, {steven.james, benjamin.rosman1}@wits.ac.za

## Abstract

The ability to compose learned skills to solve new tasks is an important property of lifelong-learning agents. In this work, we formalise the logical composition of tasks as a Boolean algebra. This allows us to formulate new tasks in terms of the negation, disjunction and conjunction of a set of base tasks. We then show that by learning goal-oriented value functions and restricting the transition dynamics of the tasks, an agent can solve these new tasks with no further learning. We prove that by composing these value functions in specific ways, we immediately recover the optimal policies for all tasks expressible under the Boolean algebra. We verify our approach in two domains—including a high-dimensional video game environment requiring function approximation—where an agent first learns a set of base skills, and then composes them to solve a super-exponential number of new tasks.

## 1 Introduction

Reinforcement learning (RL) has achieved recent success in a number of difficult, high-dimensional environments (Mnih et al., 2015; Levine et al., 2016; Lillicrap et al., 2016; Silver et al., 2017). However, these methods generally require millions of samples from the environment to learn optimal behaviours, limiting their real-world applicability. A major challenge is thus in designing sample-efficient agents that can transfer their existing knowledge to solve new tasks quickly. This is particularly important for agents in a multitask or lifelong setting, since learning to solve complex tasks from scratch is typically impractical.

One approach to transfer is *composition* (Todorov, 2009), which allows an agent to leverage existing skills to build complex, novel behaviours. These newly-formed skills can then be used to solve or speed up learning in a new task. In this work, we focus on concurrent composition, where existing base skills are combined to produce new skills (Todorov, 2009; Saxe et al., 2017; Haarnoja et al., 2018; Van Niekerk et al., 2019; Hunt et al., 2019; Peng et al., 2019). This differs from other forms of composition, such as options (Sutton et al., 1999) and hierarchical RL (Barto & Mahadevan, 2003), where actions and skills are chained in a temporal sequence.

While previous work on logical composition considers only the union and intersection of tasks (Haarnoja et al., 2018; Van Niekerk et al., 2019; Hunt et al., 2019), they do not formally define them. However, union and intersection are operations on sets, rather than tasks. We therefore formalise the notion of union and intersection of tasks using the Boolean algebra structure, since this is the algebraic structure that abstracts the notions of union, intersection, and complement of sets. We then define a Boolean algebra over the space of optimal value functions, and then prove that there exists a homomorphism between the task and value function algebras. Given a set of base tasks that have been previously solved by the agent, any new task written as a Boolean expression can immediately be solved without further learning, resulting in a zero-shot super-exponential explosion in the agent's abilities. We summarise our main contributions as follows:

1. *Boolean task algebra:* We formalise the disjunction, conjunction, and negation of tasks in a Boolean algebra structure. This extends previous composition work to encompass *all* Boolean operators, and enables us to apply logic to tasks, much as we would to propositions.

2. *Extended value functions:* We introduce a new type of goal-oriented value function that encodes how to achieve all goals in an environment. We then prove that this richer value function allows us to achieve zero-shot composition when an agent is given a new task.

3. *Zero-shot composition:* We improve on previous work (Van Niekerk et al., 2019) by showing zero-shot logical composition of tasks without any additional assumptions. This is an important result as it enables lifelong-learning agents to solve a super-exponentially increasing number of tasks as the number of base tasks they learn increase.

We illustrate our approach in the Four Rooms domain (Sutton et al., 1999), where an agent first learns to reach a number of rooms, after which it can then optimally solve any task expressible in the Boolean algebra. We then demonstrate composition in a high-dimensional video game environment, where an agent first learns to collect different objects, and then composes these abilities to solve complex tasks immediately. Our results show that, even when function approximation is required, an agent can leverage its existing skills to solve new tasks without further learning.

## 2 Preliminaries

We consider tasks modelled by Markov Decision Processes (MDPs). An MDP is defined by the tuple $(\mathcal{S}, \mathcal{A}, \rho, r)$, where (i) $\mathcal{S}$ is the state space, (ii) $\mathcal{A}$ is the action space, (iii) $\rho$ is a Markov transition kernel $(s, a) \mapsto \rho_{(s,a)}$ from $\mathcal{S} \times \mathcal{A}$ to $\mathcal{S}$, and (iv) $r$ is the real-valued reward function bounded by $[r_{\mathrm{MIN}}, r_{\mathrm{MAX}}]$. In this work, we focus on stochastic shortest path problems (Bertsekas & Tsitsiklis, 1991), which model tasks in which an agent must reach some goal. We therefore consider the class of undiscounted MDPs with an absorbing set $\mathcal{G} \subseteq \mathcal{S}$.

The goal of the agent is to compute a Markov policy $\pi$ from $\mathcal{S}$ to $\mathcal{A}$ that optimally solves a given task. A given policy $\pi$ induces a value function $V^{\pi}(s) = \mathbb{E}_{\pi} \left[ \sum_{t=0}^{\infty} r(s_t, a_t) \right]$, specifying the expected return obtained under $\pi$ starting from state $s$.[1] The *optimal* policy $\pi^*$ is the policy that obtains the greatest expected return at each state: $V^{\pi^*}(s) = V^*(s) = \max_{\pi} V^{\pi}(s)$ for all $s \in \mathcal{S}$. A related quantity is the $Q$-value function, $Q^{\pi}(s, a)$, which defines the expected return obtained by executing $a$ from $s$, and thereafter following $\pi$. Similarly, the optimal $Q$-value function is given by $Q^*(s, a) = \max_{\pi} Q^{\pi}(s, a)$ for all $(s, a) \in \mathcal{S} \times \mathcal{A}$. Finally, we denote a *proper policy* to be a policy that is guaranteed to eventually reach the absorbing set $\mathcal{G}$ (James & Collins, 2006; Van Niekerk et al., 2019). We assume the value functions for improper policies—those that never reach absorbing states—are unbounded from below.

## 3 Boolean Algebras for Tasks and Value Functions

In this section, we develop the notion of a Boolean task algebra. This formalises the notion of task conjunction ($\wedge$) and disjunction ($\vee$) introduced in previous work (Haarnoja et al., 2018; Van Niekerk et al., 2019; Hunt et al., 2019), while additionally introducing the concept of negation ($\neg$). We then show that, having solved a series of base tasks, an agent can use its knowledge to solve tasks expressible as a Boolean expression over those tasks, without any further learning.[2]

We consider a family of related MDPs $\mathcal{M}$ restricted by the following assumption:

**Assumption 1** (Van Niekerk et al. (2019)). *For all tasks in a set of tasks $\mathcal{M}$, (i) the tasks share the same state space, action space and transition dynamics, (ii) the transition dynamics are deterministic, and (iii) the reward functions between tasks differ only on the absorbing set $\mathcal{G}$. For all non-terminal states, we denote the reward $r_{s,a}$ to emphasise that it is constant across tasks.*

Assumption 1 represents the family of tasks where the environment remains the same but the goals and their desirability may vary. This is typically true for robotic navigation and manipulation tasks

where there are multiple achievable goals, the goals we want the robot to achieve may vary, and how desirable those goals are may also vary. Although we have placed restrictions on the reward functions, the above formulation still allows for a large number of tasks to be represented. Importantly, sparse rewards can be formulated under these restrictions. In practice, however, all of these assumptions can be violated with minimal impact. In particular, additional experiments in the supplementary material show that even for tasks with stochastic transition dynamics and dense rewards, and which differ in their terminal states, our composition approach still results in policies that are either identical or very close to optimal.

## 3.1 A Boolean Algebra for Tasks

An abstract Boolean algebra is a set $\mathcal{B}$ equipped with operators $\neg, \vee, \wedge$ that satisfy the Boolean axioms of (i) idempotence, (ii) commutativity, (iii) associativity, (iv) absorption, (v) distributivity, (vi) identity, and (vii) complements.[3]

We first define the $\neg, \vee,$ and $\wedge$ operators over a set of tasks.

**Definition 1.** *Let $\mathcal{M}$ be a set of tasks which adhere to Assumption 1, with $\mathcal{M}_{\mathcal{U}}, \mathcal{M}_{\varnothing} \in \mathcal{M}$ such that*

$$r_{\mathcal{M}_{\mathcal{U}}} : \mathcal{S} \times \mathcal{A} \to \mathbb{R} \qquad\qquad r_{\mathcal{M}_{\varnothing}} : \mathcal{S} \times \mathcal{A} \to \mathbb{R}$$
$$(s, a) \mapsto \max_{M \in \mathcal{M}} r_M(s, a) \qquad\qquad (s, a) \mapsto \min_{M \in \mathcal{M}} r_M(s, a)$$

*Define the $\neg, \vee,$ and $\wedge$ operators over $\mathcal{M}$ as*

$$\neg : \mathcal{M} \to \mathcal{M}$$
$$M \mapsto (\mathcal{S}, \mathcal{A}, \rho, r_{\neg M}), \text{ where } r_{\neg M} : \mathcal{S} \times \mathcal{A} \to \mathbb{R}$$
$$(s, a) \mapsto \big(r_{\mathcal{M}_{\mathcal{U}}}(s, a) + r_{\mathcal{M}_{\varnothing}}(s, a)\big) - r_M(s, a)$$

$$\vee : \mathcal{M} \times \mathcal{M} \to \mathcal{M}$$
$$(M_1, M_2) \mapsto (\mathcal{S}, \mathcal{A}, \rho, r_{M_1 \vee M_2}), \text{ where } r_{M_1 \vee M_2} : \mathcal{S} \times \mathcal{A} \to \mathbb{R}$$
$$(s, a) \mapsto \max\{r_{M_1}(s, a), r_{M_2}(s, a)\}$$

$$\wedge : \mathcal{M} \times \mathcal{M} \to \mathcal{M}$$
$$(M_1, M_2) \mapsto (\mathcal{S}, \mathcal{A}, \rho, r_{M_1 \wedge M_2}), \text{ where } r_{M_1 \wedge M_2} : \mathcal{S} \times \mathcal{A} \to \mathbb{R}$$
$$(s, a) \mapsto \min\{r_{M_1}(s, a), r_{M_2}(s, a)\}$$

In order to formalise the logical composition of tasks under the Boolean algebra structure, it is necessary that the tasks have a Boolean nature. This is enforced by the following sparseness assumption:

**Assumption 2.** *For all tasks in a set of tasks $\mathcal{M}$ which adhere to Assumption 1, the set of possible terminal rewards consists of only two values. That is, for all $(g, a)$ in $\mathcal{G} \times \mathcal{A}$, we have that $r(g, a) \in \{r_{\varnothing}, r_{\mathcal{U}}\} \subset [r_{MIN}, r_{MAX}]$ with $r_{\varnothing} \leq r_{\mathcal{U}}$.[4]*

Given the above definitions and the restrictions placed on the set of tasks we consider, we can now define a Boolean algebra over a set of tasks.

**Theorem 1.** *Let $\mathcal{M}$ be a set of tasks which adhere to Assumption 2. Then $(\mathcal{M}, \vee, \wedge, \neg, \mathcal{M}_{\mathcal{U}}, \mathcal{M}_{\varnothing})$ is a Boolean algebra.*

Theorem 1 allows us to compose existing tasks together to create new tasks in a principled way. Figure 1 illustrates the semantics for each of the Boolean operators in a simple environment.

## 3.2 Extended Value Functions

The reward and value functions described in Section 2 are insufficient to solve tasks specified by the Boolean algebra above. To understand why, consider two tasks that have multiple different goals, but at least one common goal. Clearly, there is a meaningful conjunction between them—namely, achieving the common goal. Now consider an agent that learns standard value functions for both

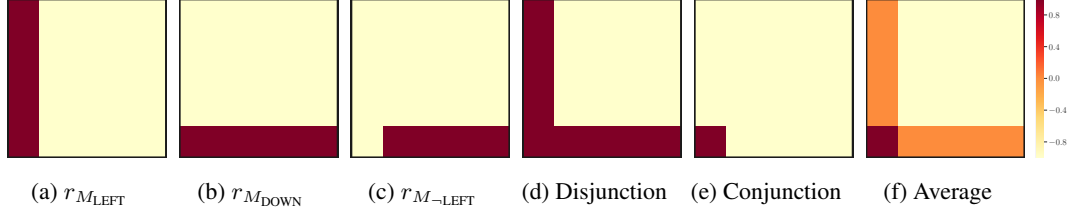

| (a) $r_{M_{\text{LEFT}}}$ | (b) $r_{M_{\text{DOWN}}}$ | (c) $r_{M_{\neg\text{LEFT}}}$ | (d) Disjunction | (e) Conjunction | (f) Average |

Figure 1: Consider two tasks, $M_{\text{LEFT}}$ and $M_{\text{DOWN}}$, in which an agent must navigate to the left and bottom regions of an $xy$-plane respectively. From left to right we plot the reward for entering a region of the state space for the individual tasks, the negation of $M_{\text{LEFT}}$, and the union (disjunction) and intersection (conjunction) of tasks. For reference, we also plot the average reward function, which has been used in previous work to approximate the conjunction operator (Haarnoja et al., 2018; Hunt et al., 2019; Van Niekerk et al., 2019). Note that by averaging reward, terminal states that are not in the intersection are erroneously given rewards.

tasks, and which is then required to solve their conjunction without further learning. Note that this is impossible in general, since the regular value function for each task only represents the value of each state with respect to the *nearest* goal. That is, for all states where the nearest goal for each task is *not* the common goal, the agent has no information about that common goal. We therefore define extended versions of the reward and value function such that the agent is able to learn the value of achieving all goals, and not simply the nearest one. These are given by the following two definitions:

**Definition 2.** *The extended reward function $\bar{r} : \mathcal{S} \times \mathcal{G} \times \mathcal{A} \to \mathbb{R}$ is given by the mapping*

$$(s, g, a) \mapsto \begin{cases} \bar{r}_{MIN} & \text{if } g \neq s \in \mathcal{G} \\ r(s, a) & \text{otherwise,} \end{cases} \tag{1}$$

*where $\bar{r}_{MIN} \leq \min\{r_{MIN}, (r_{MIN} - r_{MAX})D\}$, and $D$ is the diameter of the MDP (Jaksch et al., 2010).*[5]

Because we require that tasks share the same transition dynamics, we also require that the absorbing set of states is shared. Thus the extended reward function adds the extra constraint that, if the agent enters a terminal state for a *different* task, it should receive the largest penalty possible. In practice, we can simply set $\bar{r}_{\text{MIN}}$ to be the lowest finite value representable by the data type used for the value function.

**Definition 3.** *The extended Q-value function $\bar{Q} : \mathcal{S} \times \mathcal{G} \times \mathcal{A} \to \mathbb{R}$ is given by the mapping*

$$(s, g, a) \mapsto \bar{r}(s, g, a) + \int_{\mathcal{S}} \bar{V}^{\bar{\pi}}(s', g)\rho_{(s,a)}(ds'), \tag{2}$$

where $\bar{V}^{\bar{\pi}}(s, g) = \mathbb{E}_{\bar{\pi}} \left[ \sum_{t=0}^{\infty} \bar{r}(s_t, g, a_t) \right]$.

The extended Q-value function is similar to DG functions (Kaelbling, 1993) which also learn how to achieve all goals, except here we use task-dependent reward functions as opposed to measuring distance between states. Veeriah et al. (2018) refers to this idea of learning to achieve all goals in an environment as "mastery". We can see that the definition of extended Q-value functions encapsulates this notion for arbitrary task rewards.

The standard reward functions and value functions can be recovered from their extended versions through the following lemma.

**Lemma 1.** *Let $r_M, \bar{r}_M, Q_M^*, \bar{Q}_M^*$ be the reward function, extended reward function, optimal Q-value function, and optimal extended Q-value function for a task $M$ in $\mathcal{M}$. Then for all $(s, a)$ in $\mathcal{S} \times \mathcal{A}$, we have (i) $r_M(s, a) = \max_{g \in \mathcal{G}} \bar{r}_M(s, g, a)$, and (ii) $Q_M^*(s, a) = \max_{g \in \mathcal{G}} \bar{Q}_M^*(s, g, a)$.*

In the same way, we can also recover the optimal policy from these extended value functions by first applying Lemma 1, and acting greedily with respect to the resulting value function.

**Lemma 2.** *Denote $\mathcal{S}^- = \mathcal{S} \setminus \mathcal{G}$ as the non-terminal states of $\mathcal{M}$. Let $M_1, M_2 \in \mathcal{M}$, and let each $g$ in $\mathcal{G}$ define MDPs $M_{1,g}$ and $M_{2,g}$ with reward functions*

$$r_{M_{1,g}} := \bar{r}_{M_1}(s, g, a) \text{ and } r_{M_{2,g}} := \bar{r}_{M_2}(s, g, a) \text{ for all } (s, a) \text{ in } \mathcal{S} \times \mathcal{A}.$$

*Then for all $g$ in $\mathcal{G}$ and $s$ in $\mathcal{S}^-$,*

$$\pi_g^*(s) \in \arg\max_{a \in \mathcal{A}} Q_{M_{1,g}}^*(s, a) \text{ iff } \pi_g^*(s) \in \arg\max_{a \in \mathcal{A}} Q_{M_{2,g}}^*(s, a).$$

Combining Lemmas 1 and 2, we can extract the greedy action from the extended value function by first maximising over goals, and then selecting the maximising action: $\pi^*(s) \in \arg\max_{a \in \mathcal{A}} \max_{g \in \mathcal{G}} \bar{Q}^*(s, g, a)$. If we consider the extended value function to be a set of standard value functions (one for each goal), then this is equivalent to first performing generalised policy improvement (Barreto et al., 2017), and then selecting the greedy action.

Finally, much like the regular definition of value functions, the extended Q-value function can be written as the sum of rewards received by the agent until first encountering a terminal state.

**Corollary 1.** *Denote $G_{s:g,a}^*$ as the sum of rewards starting from $s$ and taking action $a$ up until, but not including, $g$. Then let $M \in \mathcal{M}$ and $\bar{Q}_M^*$ be the extended Q-value function. Then for all $s \in \mathcal{S}, g \in \mathcal{G}, a \in \mathcal{A}$, there exists a $G_{s:g,a}^* \in \mathbb{R}$ such that*

$$\bar{Q}_M^*(s, g, a) = G_{s:g,a}^* + \bar{r}_M(s', g, a'), \text{ where } s' \in \mathcal{G} \text{ and } a' = \arg\max_{b \in \mathcal{A}} \bar{r}_M(s', g, b).$$

### 3.3 A Boolean Algebra for Value Functions

In the same manner we constructed a Boolean algebra over a set of tasks, we can also do so for a set of optimal extended Q-value functions for the corresponding tasks.

**Definition 4.** *Let $\bar{\mathcal{Q}}^*$ be the set of optimal extended $\bar{Q}$-value functions for tasks in $\mathcal{M}$ which adhere to Assumption 1, with $\bar{Q}_\varnothing^*, \bar{Q}_{\mathcal{U}}^* \in \bar{\mathcal{Q}}^*$ the optimal $\bar{Q}$-functions for the tasks $\mathcal{M}_\varnothing, \mathcal{M}_{\mathcal{U}} \in \mathcal{M}$. Define the $\neg, \vee,$ and $\wedge$ operators over $\bar{\mathcal{Q}}^*$ as,*

$$\neg : \bar{\mathcal{Q}}^* \to \bar{\mathcal{Q}}^*$$
$$\bar{Q}^* \mapsto \neg \bar{Q}^*, \text{ where } \neg \bar{Q}^* : \mathcal{S} \times \mathcal{G} \times \mathcal{A} \to \mathbb{R}$$
$$(s, g, a) \mapsto \left( \bar{Q}_{\mathcal{U}}^*(s, g, a) + \bar{Q}_\varnothing^*(s, g, a) \right) - \bar{Q}^*(s, g, a)$$

$$\vee : \bar{\mathcal{Q}}^* \times \bar{\mathcal{Q}}^* \to \bar{\mathcal{Q}}^*$$
$$(\bar{Q}_1^*, \bar{Q}_2^*) \mapsto \bar{Q}_1^* \vee \bar{Q}_2^*, \text{ where } \bar{Q}_1^* \vee \bar{Q}_2^* : \mathcal{S} \times \mathcal{G} \times \mathcal{A} \to \mathbb{R}$$
$$(s, g, a) \mapsto \max\{\bar{Q}_1^*(s, g, a), \bar{Q}_2^*(s, g, a)\}$$

$$\wedge : \bar{\mathcal{Q}}^* \times \bar{\mathcal{Q}}^* \to \bar{\mathcal{Q}}^*$$
$$(\bar{Q}_1^*, \bar{Q}_2^*) \mapsto \bar{Q}_1^* \wedge \bar{Q}_2^*, \text{ where } \bar{Q}_1^* \wedge \bar{Q}_2^* : \mathcal{S} \times \mathcal{G} \times \mathcal{A} \to \mathbb{R}$$
$$(s, g, a) \mapsto \min\{\bar{Q}_1^*(s, g, a), \bar{Q}_2^*(s, g, a)\}$$

**Theorem 2.** *Let $\bar{\mathcal{Q}}^*$ be the set of optimal extended $\bar{Q}$-value functions for tasks in $\mathcal{M}$ which adhere to Assumption 2. Then $(\bar{\mathcal{Q}}^*, \vee, \wedge, \neg, \bar{Q}_{\mathcal{U}}^*, \bar{Q}_\varnothing^*)$ is a Boolean Algebra.*

### 3.4 Between Task and Value Function Algebras

Having established a Boolean algebra over tasks and extended value functions, we finally show that there exists an equivalence between the two. As a result, if we can write down a task under the Boolean algebra, we can immediately write down the optimal value function for the task.

**Theorem 3.** *Let $\bar{\mathcal{Q}}^*$ be the set of optimal extended $\bar{Q}$-value functions for tasks in $\mathcal{M}$ which adhere to Assumption 1. Then for all $M_1, M_2 \in \mathcal{M}$, we have (i) $\bar{Q}_{\neg M_1}^* = \neg \bar{Q}_{M_1}^*$, (ii) $\bar{Q}_{M_1 \vee M_2}^* = \bar{Q}_{M_1}^* \vee \bar{Q}_{M_2}^*$, and (iii) $\bar{Q}_{M_1 \wedge M_2}^* = \bar{Q}_{M_1}^* \wedge \bar{Q}_{M_2}^*$.*

**Corollary 2.** *Let $\mathcal{F} : \mathcal{M} \to \bar{\mathcal{Q}}^*$ be any map from $\mathcal{M}$ to $\bar{\mathcal{Q}}^*$ such that $\mathcal{F}(M) = \bar{Q}_M^*$ for all $M$ in $\mathcal{M}$. Then $\mathcal{F}$ is a homomorphism between $(\mathcal{M}, \vee, \wedge, \neg, \mathcal{M}_{\mathcal{U}}, \mathcal{M}_\varnothing)$ and $(\bar{\mathcal{Q}}^*, \vee, \wedge, \neg, \bar{Q}_{\mathcal{U}}^*, \bar{Q}_\varnothing^*)$.*

Theorem 3 shows that we can provably achieve zero-shot negation, disjunction, and conjunction provided Assumption 1 is satisfied. Corollary 2 extends this result by showing that the task and value function algebras are in fact homomorphic, which implies zero-shot composition of arbitrary combinations of negations, disjunctions, and conjunctions.

# 4 Zero-shot Transfer Through Composition

We can use the theory developed in the previous sections to perform zero-shot transfer by first learning extended value functions for a set of base tasks, and then composing them to solve new tasks expressible under the Boolean algebra. To demonstrate this, we conduct a series of experiments in the Four Rooms domain (Sutton et al., 1999), where an agent must navigate a grid world to a particular location. The agent can move in any of the four cardinal directions at each timestep, but colliding with a wall leaves the agent in the same location. We add a 5th action for "stay" that the agent chooses to achieve goals. A goal position only becomes terminal if the agent chooses to stay in it. The transition dynamics are deterministic, and rewards are $-0.1$ for all non-terminal states, and $2$ at the goal.

## 4.1 Learning Base Tasks

We use a modified version of Q-learning (Watkins, 1989) to learn the extended Q-value functions described previously. Our algorithm differs in a number of ways from standard Q-learning: we keep track of the set of terminating states seen so far, and at each timestep we update the extended Q-value function with respect to both the current state and action, as well as all goals encountered so far. We also use the definition of the extended reward function, and so if the agent encounters a terminal state of a different task, it receives reward $\bar{r}_{\mathrm{MIN}}$. The full pseudocode is listed in the supplementary material.

If we know the set of goals (and hence potential base tasks) upfront, then it is easy to select a minimal set of base tasks that can be composed to produce the largest number of composite tasks. We first assign a Boolean label to each goal in a table, and then use the columns of the table as base tasks. The goals for each base task are then those goals with value $1$ according to the table. In this domain, the two base tasks we select are $M_{\mathrm{T}}$, which requires that the agent visit either of the top two rooms, and $M_{\mathrm{L}}$, which requires visiting the two left rooms. We illustrate this selection procedure in the supplementary material.

## 4.2 Boolean Composition

Having learned the optimal extended value functions for our base tasks, we can now leverage Theorems 1–3 to solve new tasks with no further learning. Figure 2 illustrates this composition, where an agent is able to immediately solve complex tasks such as exclusive-or. We illustrate a few composite tasks here, but note that in general, if we have $K$ base tasks, then a Boolean algebra allows for $2^{2^K}$ new tasks to be constructed. Thus having trained on only two tasks, our agent has enough information to solve a total of 16 composite tasks.

By learning extended value functions, an agent can subsequently solve a massive number of tasks; however, the upfront cost of learning is likely to be higher. We investigate the trade-off between the two approaches by quantifying how the sample complexity scales with the number of tasks. We compare to Van Niekerk et al. (2019), who use regular value functions to demonstrate optimal disjunctive composition. We note that while the upfront learning cost is therefore lower, the number of tasks expressible using only disjunction is $2^K - 1$, which is significantly less than the full Boolean algebra. We also conduct a test using an extended version of the Four Rooms domain, where additional goals are placed along the sides of all walls, resulting in a total of 40 goals. Empirical results are illustrated by Figure 3.

Our results show that while additional samples are needed to learn an extended value function, the agent is able to expand the tasks it can solve super-exponentially. Furthermore, the number of base tasks we need to solve is only logarithmic in the number of goal states. For an environment with $K$ goals, we need to learn only $\lfloor \log_2 K \rfloor + 1$ base tasks, as opposed to the disjunctive approach which requires $K$ base tasks. Thus by sacrificing sample efficiency initially, we achieve an exponential increase in abilities compared to previous work (Van Niekerk et al., 2019).

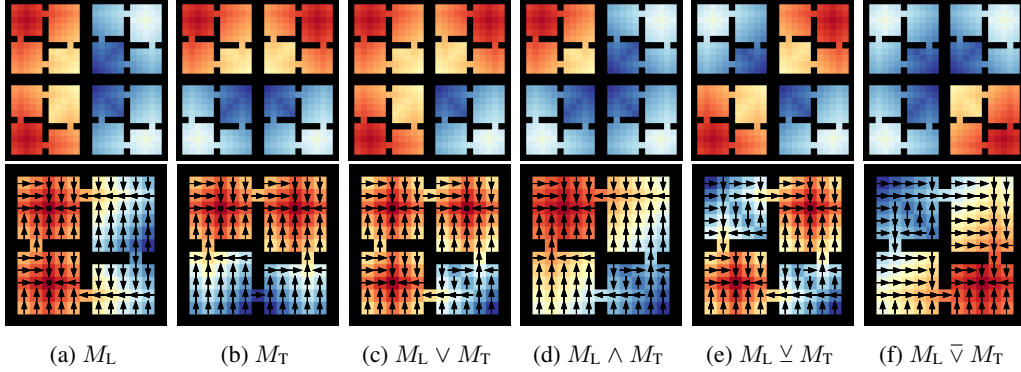

(a) $M_L$      (b) $M_T$      (c) $M_L \vee M_T$      (d) $M_L \wedge M_T$      (e) $M_L \veebar M_T$      (f) $M_L \, \overline{\vee} \, M_T$

Figure 2: An example of zero-shot Boolean algebraic composition using the learned extended value functions. The top row shows the extended value functions. For each, the plots show the value of each state with respect to the four goals (the centre of each room). The bottom row shows the recovered regular value functions obtained by maximising over goals. Arrows represent the optimal action in a given state. (a–b) The learned optimal extended value functions for the base tasks. (c) Zero-shot disjunctive composition. (d) Zero-shot conjunctive composition. (e) Combining operators to model exclusive-or composition. (f) Composition that produces logical nor. Note that the resulting optimal value function can attain a goal not explicitly represented by the base tasks.

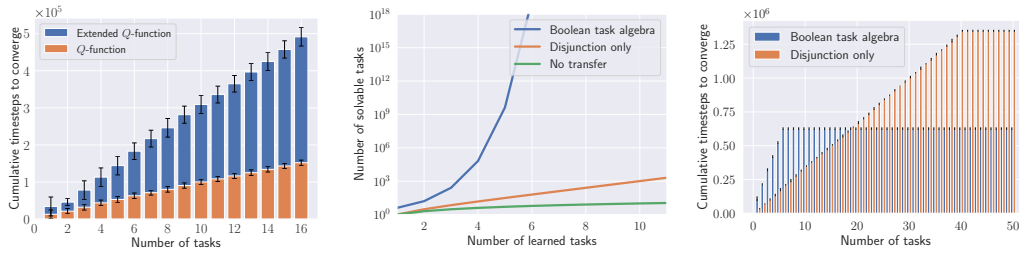

(a) Cumulative number of samples required to learn optimal extended and regular value functions. Error bars represent standard deviations over 100 runs.

(b) Number of tasks that can be solved as a function of the number of existing tasks solved. Results are plotted on a log-scale.

(c) Cumulative number of samples required to solve tasks in a 40-goal Four Rooms domain. Error bars represent standard deviations over 100 runs.

Figure 3: Results in comparison to the disjunctive composition of Van Niekerk et al. (2019). (a) The number of samples required to learn the extended value function is greater than learning a standard value function. However, both scale linearly and differ only by a constant factor. (b) The extended value functions allow us to solve exponentially more tasks than the disjunctive approach without further learning. (c) In the modified task with 40 goals, we need to learn only 7 base tasks, as opposed to 40 for the disjunctive case.

## 5    Composition with Function Approximation

Finally, we demonstrate that our compositional approach can also be used to tackle high-dimensional domains where function approximation is required. We use the same video game environment as Van Niekerk et al. (2019), where an agent must navigate a 2D world and collect objects of different shapes and colours from any initial position. The state space is an $84 \times 84$ RGB image, and the agent is able to move in any of the four cardinal directions. The agent also possesses a `pick-up` action, which allows it to collect an object when standing on top of it. There are two shapes (squares and circles) and three colours (blue, beige and purple) for a total of six unique objects.

To learn the extended action-value functions, we modify deep Q-learning (Mnih et al., 2015) similarly to the many-goals update method of Veeriah et al. (2018). Here, a universal value function approximator (UVFA) (Schaul et al., 2015) is used to represent the action values for each state and

goal (both specified as RGB images).[6] Additionally, when a terminal state is encountered, it is added to the collection of goals seen so far, and when learning updates occur, these goals are sampled randomly from a replay buffer. We first learn to solve two base tasks: collecting blue objects and collecting squares. As shown in Figure 4, by training the UVFA for each task using the extended rewards definition, the agent learns not only how to achieve all goals, but also how desirable each of those goals are for the current task. These UVFAs can now be composed to solve new tasks with no further learning.

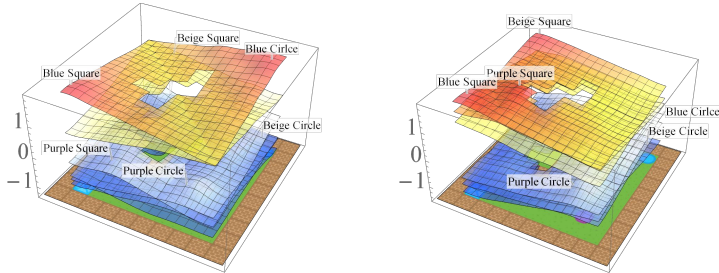

Figure 4: Extended value function for collecting blue objects (left) and squares (right). To generate the value functions, we place the agent at every location and compute the maximum output of the network over all goals and actions. We then interpolate between the points to smooth the graph. Any error in the visualisation is due to the use of non-linear function approximation.

We demonstrate composition characterised by disjunction, conjunction and exclusive-or. This corresponds to tasks where the target items are: (i) blue or square, (ii) blue squares, and (iii) blue or squares, but not blue squares. Figure 5 illustrates the composed value functions and samples of the subsequent trajectories for the respective tasks. Figure 6 shows the average returns across random initial positions of the agent.[7]

## 6  Related Work

The ability to compose value functions was first demonstrated using the linearly-solvable MDP framework (Todorov, 2007), where value functions could be composed to solve tasks similar to the disjunctive case (Todorov, 2009). Van Niekerk et al. (2019) show that the same kind of composition can be achieved using entropy-regularised RL (Fox et al., 2016), and extend the results to the standard RL setting, where agents can optimally solve the disjunctive case. Using entropy-regularised RL, Haarnoja et al. (2018) approximates the conjunction of tasks by averaging their reward functions, and demonstrates that by averaging the optimal value functions of the respective tasks, the agent can achieve performance close to optimal. Hunt et al. (2019) extends this result by composing value functions to solve the average reward task exactly, which approximates the true conjunctive case. More recently, Peng et al. (2019) introduce a few-shot learning approach to compose policies multiplicatively. Although lacking theoretical foundations, their results show that an agent can learn a weighted composition of existing base skills to solve a new complex task. By contrast, we show that zero-shot optimal composition can be achieved for all Boolean operators.

## 7  Conclusion

We have shown how to compose tasks using the standard Boolean algebra operators. These composite tasks can be solved without further learning by first learning goal-oriented value functions, and then composing them in a similar manner. Finally, we note that there is much room for improvement in learning the extended value functions for the base tasks. In our experiments, we learned each extended value function from scratch, but it is likely that having learned one for the first task, we could use it to initialise the extended value function for the second task to improve convergence times. One area for improvement lies in efficiently learning the extended value functions, as well as developing better algorithms for solving tasks with sparse rewards. For example, it is likely that

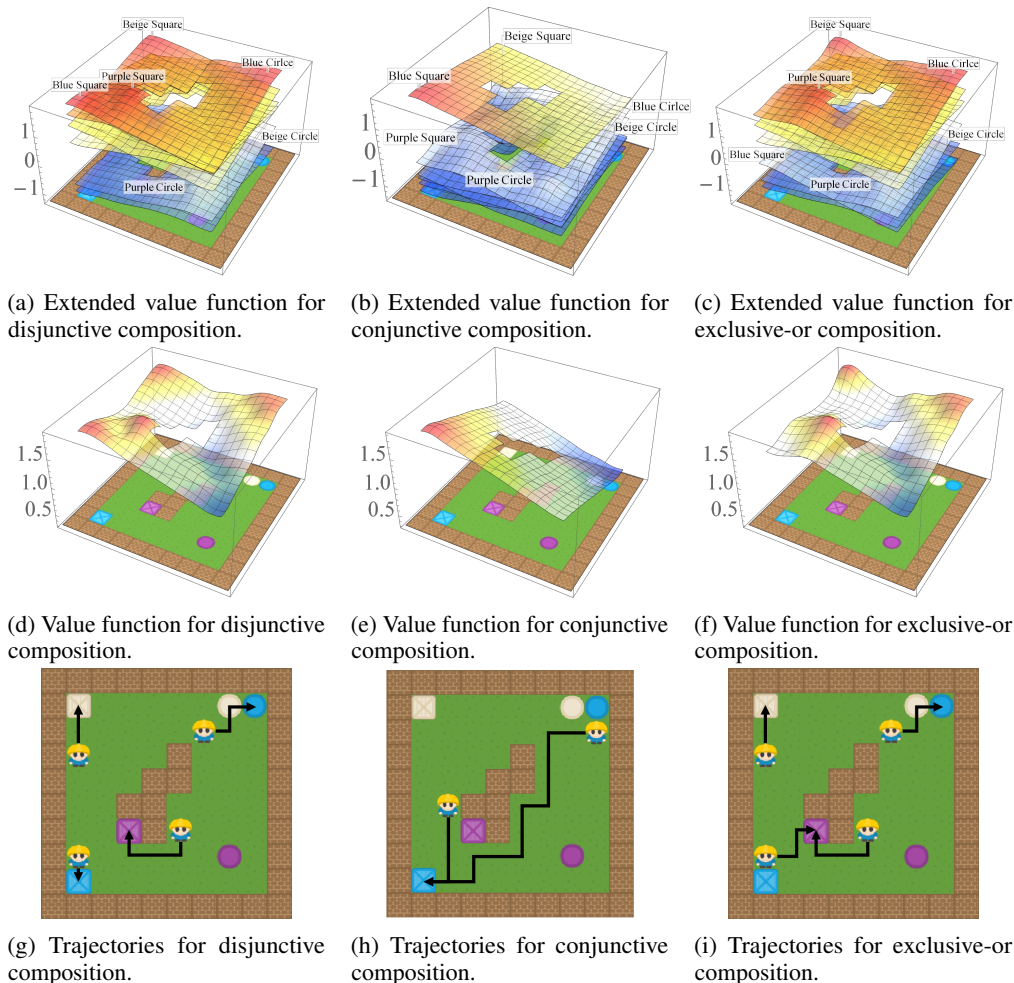

(a) Extended value function for disjunctive composition.

(b) Extended value function for conjunctive composition.

(c) Extended value function for exclusive-or composition.

(d) Value function for disjunctive composition.

(e) Value function for conjunctive composition.

(f) Value function for exclusive-or composition.

(g) Trajectories for disjunctive composition.

(h) Trajectories for conjunctive composition.

(i) Trajectories for exclusive-or composition.

Figure 5: By composing extended value functions from the base tasks (collecting blue objects, and collecting squares), we can act optimally in new tasks with no further learning.

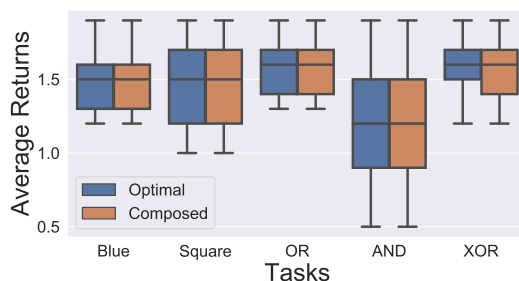

Figure 6: Average returns over 1000 episodes for the *Blue* and *Square* tasks, and their disjunction (*OR*), conjunction (*AND*) and exlusive-or (*XOR*).

approaches such as hindsight experience replay (Andrychowicz et al., 2017) could reduce the number of samples required to learn extended value functions, while Mirowski et al. (2017) provides a method for learning complex tasks with sparse rewards using auxiliary tasks. We leave incorporating these approaches to future work, but note that our framework is agnostic to the value-function learning algorithm. Our proposed approach is a step towards both interpretable RL—since both the tasks and optimal value functions can be specified using Boolean operators—and the ultimate goal of lifelong learning agents, which are able to solve combinatorially many tasks in a sample-efficient manner.

## Broader Impact

Our work is mainly theoretical, but is a step towards creating agents that can solve tasks specified using human-understandable Boolean expressions, which could one day be deployed in practical RL systems. We envisage this as an avenue for overcoming the problem of reward misspecification, and for developing safer agents whose goals are readily interpretable by humans.

## Acknowledgments and Disclosure of Funding

The authors wish to thank the anonymous reviewers for their helpful comments, and Pieter Abbeel, Marc Deisenroth and Shakir Mohamed for their assistance in reviewing a final draft of this paper. This work is based on the research supported in part by the National Research Foundation of South Africa (Grant Number: 17808).

## Footnotes

[1]Since we consider undiscounted MDPs, we can ensure the value function is bounded by augmenting the state space with a virtual state $\omega$ such that $\rho_{(s,a)}(\omega) = 1$ for all $(s, a) \in \mathcal{G} \times \mathcal{A}$, and $r = 0$ after reaching $\omega$.

[2]Owing to space constraints, all proofs are presented in the supplementary material.

[3]We provide a description of these axioms in the supplementary material.

[4]While Assumption 2 is necessary to establish the Boolean algebra, we show in Theorem 3 that it is not required for zero-shot negation, disjunction, and conjunction.

[5]The diameter is defined as $D = \max_{s \neq s' \in \mathcal{S}} \min_{\pi} \mathbb{E}\left[T(s'|\pi, s)\right]$, where $T$ is the number of timesteps required to first reach $s'$ from $s$ under $\pi$.

[6]The hyperparameters and network architecture are listed in the supplementary material

[7]Experiments involving randomised object positions are included in the supplementary material.

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
