[Supplementary Material]

# Supplementary Material:
# A Boolean Task Algebra For Reinforcement Learning

**Geraud Nangue Tasse, Steven James, Benjamin Rosman**
School of Computer Science and Applied Mathematics
University of the Witwatersrand
Johannesburg, South Africa
geraudnt@gmail.com, {steven.james, benjamin.rosman1}@wits.ac.za

## 1 Boolean Algebra Definition

**Definition 1.** *A Boolean algebra is a set $\mathcal{B}$ equipped with the binary operators $\vee$ (disjunction) and $\wedge$ (conjunction), and the unary operator $\neg$ (negation), which satisfies the following Boolean algebra axioms for $a, b, c$ in $\mathcal{B}$:*

*(i) Idempotence: $a \wedge a = a \vee a = a$.*

*(ii) Commutativity: $a \wedge b = b \wedge a$ and $a \vee b = b \vee a$.*

*(iii) Associativity: $a \wedge (b \wedge c) = (a \wedge b) \wedge c$ and $a \wedge (b \vee c) = (a \vee b) \vee c$.*

*(iv) Absorption: $a \wedge (a \vee b) = a \vee (a \wedge b) = a$.*

*(v) Distributivity: $a \wedge (b \vee c) = (a \wedge b) \vee (a \wedge c)$ and $a \vee (b \wedge c) = (a \vee b) \wedge (a \vee c)$.*

*(vi) Identity: there exists $\mathbf{0}, \mathbf{1}$ in $\mathcal{B}$ such that*

$$\mathbf{0} \wedge a = \mathbf{0}$$
$$\mathbf{0} \vee a = a$$
$$\mathbf{1} \wedge a = a$$
$$\mathbf{1} \vee a = \mathbf{1}$$

*(vii) Complements: for every $a$ in $\mathcal{B}$, there exists an element $a'$ in $\mathcal{B}$ such that $a \wedge a' = \mathbf{0}$ and $a \vee a' = \mathbf{1}$.*

## 2 Proof for Boolean Task Algebra

**Theorem 1.** *Let $\mathcal{M}$ be a set of tasks which adhere to Assumption 2. Then $(\mathcal{M}, \vee, \wedge, \neg, \mathcal{M}_{\mathcal{U}}, \mathcal{M}_{\varnothing})$ is a Boolean algebra.*

*Proof.* Let $M_1, M_2 \in \mathcal{M}$. We show that $\neg, \vee, \wedge$ satisfy the Boolean properties (i) – (vii).

**(i)–(v):** These easily follow from the fact that the $\min$ and $\max$ functions satisfy the idempotent, commutative, associative, absorption and distributive laws.

**(vi):** Let $r_{\mathcal{M}_\mathcal{U} \wedge M_1}$ and $r_{M_1}$ be the reward functions for $\mathcal{M}_\mathcal{U} \wedge M_1$ and $M_1$ respectively. Then for all $(s, a)$ in $\mathcal{S} \times \mathcal{A}$,

$$r_{\mathcal{M}_\mathcal{U} \wedge M_1}(s, a) = \begin{cases} \min\{r_\mathcal{U}, r_{M_1}(s,a)\}, & \text{if } s \in \mathcal{G} \\ \min\{r_{s,a}, r_{s,a}\}, & \text{otherwise.} \end{cases}$$

$$= \begin{cases} r_{M_1}(s,a), & \text{if } s \in \mathcal{G} \\ r_{s,a}, & \text{otherwise.} \end{cases} \qquad (r_{M_1}(s,a) \in \{r_\varnothing, r_\mathcal{U}\} \text{ for } s \in \mathcal{G})$$

$$= r_{M_1}(s,a).$$

Thus $\mathcal{M}_\mathcal{U} \wedge M_1 = M_1$. Similarly $\mathcal{M}_\mathcal{U} \vee M_1 = \mathcal{M}_\mathcal{U}$, $\mathcal{M}_\varnothing \wedge M_1 = \mathcal{M}_\varnothing$, and $\mathcal{M}_\varnothing \vee M_1 = M_1$. Hence $\mathcal{M}_\varnothing$ and $\mathcal{M}_\mathcal{U}$ are the universal bounds of $\mathcal{M}$.

**(vii):** Let $r_{M_1 \wedge \neg M_1}$ be the reward function for $M_1 \wedge \neg M_1$. Then for all $(s, a)$ in $\mathcal{S} \times \mathcal{A}$,

$$r_{M_1 \wedge \neg M_1}(s, a) = \begin{cases} \min\{r_{M_1}(s,a), (r_\mathcal{U} + r_\varnothing) - r_{M_1}(s,a)\}, & \text{if } s \in \mathcal{G} \\ \min\{r_{s,a}, (r_{s,a} + r_{s,a}) - r_{s,a}\}, & \text{otherwise.} \end{cases}$$

$$= \begin{cases} r_\varnothing, & \text{if } s \in \mathcal{G} \text{ and } r_{M_1}(s,a) = r_\mathcal{U} \\ r_\varnothing, & \text{if } s \in \mathcal{G} \text{ and } r_{M_1}(s,a) = r_\varnothing \\ r_{s,a}, & \text{otherwise.} \end{cases}$$

$$= r_{\mathcal{M}_\varnothing}(s, a).$$

Thus $M_1 \wedge \neg M_1 = \mathcal{M}_\varnothing$, and similarly $M_1 \vee \neg M_1 = \mathcal{M}_\mathcal{U}$.

$\square$

## 3  Proofs of Properties of Extended Value Functions

**Lemma 1.** *Let $r_M, \bar{r}_M, Q_M^*, \bar{Q}_M^*$ be the reward function, extended reward function, optimal Q-value function, and optimal extended Q-value function for a task $M$ in $\mathcal{M}$. Then for all $(s, a)$ in $\mathcal{S} \times \mathcal{A}$, we have (i) $r_M(s, a) = \max_{g \in \mathcal{G}} \bar{r}_M(s, g, a)$, and (ii) $Q_M^*(s, a) = \max_{g \in \mathcal{G}} \bar{Q}_M^*(s, g, a)$.*

*Proof.*

**(i):**

$$\max_{g \in \mathcal{G}} \bar{r}_M(s, g, a) = \begin{cases} \max\{\bar{r}_{\text{MIN}}, r_M(s,a)\}, & \text{if } s \in \mathcal{G} \\ \max_{g \in \mathcal{G}} r_M(s,a), & \text{otherwise.} \end{cases}$$

$$= r_M(s, a) \qquad\qquad (\bar{r}_{\text{MIN}} \leq r_{\text{MIN}} \leq r_M(s,a) \text{ by definition}).$$

**(ii):** Each $g$ in $\mathcal{G}$ can be thought of as defining an MDP $M_g := (\mathcal{S}, \mathcal{A}, \rho, r_{M_g})$ with reward function $r_{M_g}(s, a) := \bar{r}_M(s, g, a)$ and optimal Q-value function $Q_{M_g}^*(s, a) = \bar{Q}_M^*(s, g, a)$. Then using (i) we have $r_M(s, a) = \max_{g \in \mathcal{G}} r_{M_g}(s, a)$ and from Van Niekerk et al. (2019, Corollary 1), we have that $Q_M^*(s, a) = \max_{g \in \mathcal{G}} Q_{M_g}^*(s, a) = \max_{g \in \mathcal{G}} \bar{Q}_M^*(s, g, a)$.

$\square$

**Lemma 2.** *Denote $\mathcal{S}^- = \mathcal{S} \setminus \mathcal{G}$ as the non-terminal states of $\mathcal{M}$. Let $M_1, M_2 \in \mathcal{M}$, and let each $g$ in $\mathcal{G}$ define MDPs $M_{1,g}$ and $M_{2,g}$ with reward functions*

$$r_{M_{1,g}} := \bar{r}_{M_1}(s, g, a) \text{ and } r_{M_{2,g}} := \bar{r}_{M_2}(s, g, a) \text{ for all } (s, a) \text{ in } \mathcal{S} \times \mathcal{A}.$$

*Then for all $g$ in $\mathcal{G}$ and $s$ in $\mathcal{S}^-$,*

$$\pi_g^*(s) \in \arg\max_{a \in \mathcal{A}} Q_{M_{1,g}}^*(s, a) \text{ iff } \pi_g^*(s) \in \arg\max_{a \in \mathcal{A}} Q_{M_{2,g}}^*(s, a).$$

*Proof.* Let $g \in \mathcal{G}, s \in \mathcal{S}^-$ and let $\pi_g^*$ be defined by

$$\pi_g^*(s') \in \arg\max_{a \in \mathcal{A}} Q_{M_1,g}^*(s,a) \text{ for all } s' \in \mathcal{S}.$$

If $g$ is unreachable from $s$, then we are done since for all $(s',a)$ in $\mathcal{S} \times \mathcal{A}$ we have

$$g \neq s' \implies r_{M_1,g}(s',a) = \begin{cases} \bar{r}_{\text{MIN}}, & \text{if } s' \in \mathcal{G} \\ r_{s',a}, & \text{otherwise} \end{cases} = r_{M_2,g}(s',a)$$

$$\implies M_{1,g} = M_{2,g}.$$

If $g$ *is* reachable from $s$, then we show that following $\pi_g^*$ must reach $g$. Since $\pi_g^*$ is proper, it must reach a terminal state $g' \in \mathcal{G}$. Assume $g' \neq g$. Let $\pi_g$ be a policy that produces the shortest trajectory to $g$. Let $G^{\pi_g^*}$ and $G^{\pi_g}$ be the returns for the respective policies. Then,

$$G^{\pi_g^*} \geq G^{\pi_g}$$

$$\implies G_{T-1}^{\pi_g^*} + r_{M_1,g}(g',\pi_g^*(g')) \geq G^{\pi_g},$$

$$\text{where } G_{T-1}^{\pi_g^*} = \sum_{t=0}^{T-1} r_{M_1,g}(s_t, \pi_g^*(s_t)) \text{ and } T \text{ is the time at which } g' \text{ is reached.}$$

$$\implies G_{T-1}^{\pi_g^*} + \bar{r}_{\text{MIN}} \geq G^{\pi_g}, \text{ since } g \neq g' \in \mathcal{G}$$

$$\implies \bar{r}_{\text{MIN}} \geq G^{\pi_g} - G_{T-1}^{\pi_g^*}$$

$$\implies (r_{\text{MIN}} - r_{\text{MAX}})D \geq G^{\pi_g} - G_{T-1}^{\pi_g^*}, \text{ by definition of } \bar{r}_{\text{MIN}}$$

$$\implies G_{T-1}^{\pi_g^*} - r_{\text{MAX}}D \geq G^{\pi_g} - r_{\text{MIN}}D, \text{ since } G^{\pi_g} \geq r_{\text{MIN}}D$$

$$\implies G_{T-1}^{\pi_g^*} - r_{\text{MAX}}D \geq 0$$

$$\implies G_{T-1}^{\pi_g^*} \geq r_{\text{MAX}}D.$$

But this is a contradiction since the result obtained by following an optimal trajectory up to a terminal state without the reward for entering the terminal state must be strictly less that receiving $r_{\text{MAX}}$ for every step of the longest possible optimal trajectory. Hence we must have $g' = g$. Similarly, all optimal policies of $M_{2,g}$ must reach $g$. Hence $\pi_g^*(s) \in \arg\max_{a \in \mathcal{A}} Q_{M_2,g}^*(s,a)$. Since $M_1$ and $M_2$ are arbitrary elements of $\mathcal{M}$, the reverse implication holds too.

$\square$

**Corollary 1.** *Denote $G_{s:g,a}^*$ as the sum of rewards starting from $s$ and taking action $a$ up until, but not including, $g$. Then let $M \in \mathcal{M}$ and $\bar{Q}_M^*$ be the extended Q-value function. Then for all $s \in \mathcal{S}, g \in \mathcal{G}, a \in \mathcal{A}$, there exists a $G_{s:g,a}^* \in \mathbb{R}$ such that*

$$\bar{Q}_M^*(s,g,a) = G_{s:g,a}^* + \bar{r}_M(s',g,a'), \text{ where } s' \in \mathcal{G} \text{ and } a' = \arg\max_{b \in \mathcal{A}} \bar{r}_M(s',g,b).$$

*Proof.* This follows directly from Lemma 2. Since all tasks $M \in \mathcal{M}$ share the same optimal policy $\pi_g^*$ up to (but not including) the goal state $g \in \mathcal{G}$, their return $G_{T-1}^{\pi_g^*} = \sum_{t=0}^{T-1} r_M(s_t, \pi_g^*(s_t))$ is the same up to (but not including) $g$. $\square$

## 4   Proof for Boolean Extendend Value Functions Algebra

**Theorem 2.** *Let $\bar{\mathcal{Q}}^*$ be the set of optimal extended $\bar{Q}$-value functions for tasks in $\mathcal{M}$ which adhere to Assumption 2. Then $(\bar{\mathcal{Q}}^*, \vee, \wedge, \neg, \bar{Q}_\mathcal{U}^*, \bar{Q}_\varnothing^*)$ is a Boolean Algebra.*

*Proof.* Let $\bar{Q}_{M_1}^*, \bar{Q}_{M_2}^* \in \bar{\mathcal{Q}}^*$ be the optimal $\bar{Q}$-value functions for tasks $M_1, M_2 \in \mathcal{M}$ with reward functions $r_{M_1}$ and $r_{M_2}$. We show that $\neg, \vee, \wedge$ satisfy the Boolean properties (i) – (vii).

**(i)–(v):** These follow directly from the properties of the $\min$ and $\max$ functions.

**(vi):** For all $(s, g, a)$ in $\mathcal{S} \times \mathcal{G} \times \mathcal{A}$,

$$
\begin{aligned}
(\bar{Q}_{\mathcal{U}}^* \wedge \bar{Q}_{M_1}^*)(s, g, a) &= \min\{(\bar{Q}_{\mathcal{U}}^*(s, g, a), \bar{Q}_{M_1}^*(s, g, a)\} \\
&= \min\{G_{s:g,a}^* + \bar{r}_{\mathcal{M}_{\mathcal{U}}}(s', g, a'), G_{s:g,a}^* + \bar{r}_{M_1}(s', g, a')\} \quad \text{(Corollary 1)} \\
&= G_{s:g,a}^* + \min\{\bar{r}_{\mathcal{M}_{\mathcal{U}}}(s', g, a'), \bar{r}_{M_1}(s', g, a')\} \\
&= G_{s:g,a}^* + \bar{r}_{M_1}(s', g, a') \qquad \text{(since } \bar{r}_{M_1}(s', g, a') \in \{r_{\varnothing}, r_{\mathcal{U}}, \bar{r}_{\text{MIN}}\}) \\
&= \bar{Q}_{M_1}^*(s, g, a).
\end{aligned}
$$

Similarly, $\bar{Q}_{\mathcal{U}}^* \vee \bar{Q}_{M_1}^* = \bar{Q}_{\mathcal{U}}^*, \bar{Q}_{\varnothing}^* \wedge \bar{Q}_{M_1}^* = \bar{Q}_{\varnothing}^*$, and $\bar{Q}_{\varnothing}^* \vee \bar{Q}_{M_1}^* = \bar{Q}_{M_1}^*$.

**(vii):** For all $(s, g, a)$ in $\mathcal{S} \times \mathcal{G} \times \mathcal{A}$,

$$
\begin{aligned}
(\bar{Q}_{M_1}^* \wedge \neg \bar{Q}_{M_1}^*)(s, g, a) &= \min\{\bar{Q}_{M_1}^*(s, g, a), (\bar{Q}_{\mathcal{U}}^*(s, g, a) - \bar{Q}_{\varnothing}^*(s, g, a)) - \bar{Q}_{M_1}^*(s, g, a)\} \\
&= G_{s:g,a}^* + \min\{\bar{r}_{M_1}(s', g, a'), (\bar{r}_{\mathcal{M}_{\mathcal{U}}}(s', g, a') + \bar{r}_{\mathcal{M}_{\varnothing}}(s', g, a')) \\
&\quad - \bar{r}_{M_1}(s', g, a')\} \\
&= G_{s:g,a}^* + \bar{r}_{\mathcal{M}_{\varnothing}}(s', g, a') \\
&= \bar{Q}_{\varnothing}^*(s, g, a).
\end{aligned}
$$

Similarly, $\bar{Q}_{M_1}^* \vee \neg \bar{Q}_{M_1}^* = \bar{Q}_{\mathcal{U}}^*$.

$\square$

# 5 Proof for Zero-shot Composition

**Theorem 3.** *Let $\bar{\mathcal{Q}}^*$ be the set of optimal extended $\bar{Q}$-value functions for tasks in $\mathcal{M}$ which adhere to Assumption 1. Then for all $M_1, M_2 \in \mathcal{M}$, we have (i) $\bar{Q}_{\neg M_1}^* = \neg \bar{Q}_{M_1}^*$, (ii) $\bar{Q}_{M_1 \vee M_2}^* = \bar{Q}_{M_1}^* \vee \bar{Q}_{M_2}^*$, and (iii) $\bar{Q}_{M_1 \wedge M_2}^* = \bar{Q}_{M_1}^* \wedge \bar{Q}_{M_2}^*$.*

*Proof.* Let $M_1, M_2 \in \mathcal{M}$. Then for all $(s, g, a)$ in $\mathcal{S} \times \mathcal{G} \times \mathcal{A}$,

**(i):**

$$
\begin{aligned}
\bar{Q}_{\neg M_1}^*(s, g, a) &= G_{s:g,a}^* + \bar{r}_{\neg M_1}(s', g, a') \quad \text{(from Corollary 1)} \\
&= G_{s:g,a}^* + (\bar{r}_{\mathcal{M}_{\mathcal{U}}}(s', g, a') + \bar{r}_{\mathcal{M}_{\varnothing}}(s', g, a')) - \bar{r}_{M_1}(s', g, a') \\
&= [(G_{s:g,a}^* + \bar{r}_{\mathcal{M}_{\mathcal{U}}}(s', g, a')) + (G_{s:g,a}^* + \bar{r}_{\mathcal{M}_{\varnothing}}(s', g, a'))] - (G_{s:g,a}^* + \bar{r}_{M_1}(s', g, a')) \\
&= [\bar{Q}_{\mathcal{U}}^*(s, g, a) + \bar{Q}_{\varnothing}^*(s, g, a)] - \bar{Q}_{M_1}^*(s, g, a) \\
&= \neg \bar{Q}_{M_1}^*(s, g, a)
\end{aligned}
$$

**(ii):**

$$
\begin{aligned}
\bar{Q}_{M_1 \vee M_2}^*(s, g, a) &= G_{s:g,a}^* + \bar{r}_{M_1 \vee M_2}(s', g, a') \\
&= G_{s:g,a}^* + \max\{\bar{r}_{M_1}(s', g, a'), \bar{r}_{M_2}(s', g, a'')\} \\
&= \max\{G_{s:g,a}^* + \bar{r}_{M_1}(s', g, a'), G_{s:g,a}^* + \bar{r}_{M_2}(s', g, a'')\} \\
&= \max\{\bar{Q}_{M_1}^*(s, g, a), \bar{Q}_{M_2}^*(s, g, a)\} \\
&= (\bar{Q}_{M_1}^* \vee \bar{Q}_{M_2}^*)(s, g, a)
\end{aligned}
$$

**(iii):** Follows similarly to (ii).

$\square$

**Corollary 2.** *Let $\mathcal{F} : \mathcal{M} \to \bar{\mathcal{Q}}^*$ be any map from $\mathcal{M}$ to $\bar{\mathcal{Q}}^*$ such that $\mathcal{F}(M) = \bar{Q}_M^*$ for all $M$ in $\mathcal{M}$. Then $\mathcal{F}$ is a homomorphism between $(\mathcal{M}, \vee, \wedge, \neg, \mathcal{M}_{\mathcal{U}}, \mathcal{M}_{\varnothing})$ and $(\bar{\mathcal{Q}}^*, \vee, \wedge, \neg, \bar{Q}_{\mathcal{U}}^*, \bar{Q}_{\varnothing}^*)$.*

*Proof.* This follows from Theorem 3. $\square$

# 6   Goal-oriented Q-learning

Below we list the pseudocode for the modified Q-learning algorithm used in the four-rooms domain.

---

**Algorithm 1:** Goal-oriented $Q$-learning

**Input :** Learning rate $\alpha$, discount factor $\gamma$, exploration constant $\varepsilon$, lower-bound extended reward
$\bar{r}_{\text{MIN}}$

Initialise $Q : \mathcal{S} \times \mathcal{S} \times \mathcal{A} \to \mathbb{R}$ arbitrarily
$\mathcal{G} \leftarrow \varnothing$
**while** *Q is not converged* **do**
    Initialise state $s$
    **while** *s is not terminal* **do**
        **if** $\mathcal{G} = \varnothing$ **then**
            Select random action $a$
        **else**

$$a \leftarrow \begin{cases} \underset{b \in \mathcal{A}}{\arg\max} \left( \underset{t \in \mathcal{G}}{\max} Q(s, t, b) \right) & \text{with probability } 1 - \varepsilon \\ \text{a random action} & \text{with probability } \varepsilon \end{cases}$$

        **end**
        Choose $a$ from $s$ according to policy derived from $Q$
        Take action $a$, observe $r$ and $s'$
        **foreach** $g \in \mathcal{G}$ **do**
            **if** $s'$ *is terminal* **then**
                **if** $s' \neq g$ **then**
                    $\delta \leftarrow \bar{r}_{\text{MIN}}$
                **else**
                    $\delta \leftarrow r - Q(s, g, a)$
                **end**
            **else**
                $\delta \leftarrow r + \gamma \max_b Q(s', g, b) - Q(s, g, a)$
            **end**
            $Q(s, g, a) \leftarrow Q(s, g, a) + \alpha\delta$
        **end**
        $s \leftarrow s'$
    **end**
    $\mathcal{G} \leftarrow \mathcal{G} \cup \{s\}$
**end**
**return** $Q$

---

Figure 1: A $Q$-learning algorithm for learning extended value functions. Note that the greedy action selection step is equivalent to generalised policy improvement (Barreto et al., 2017) over the set of extended value functions.

# 7 Investigating Practical Considerations

The theoretical results presented in this work rely on Assumptions 1 and 2, which restrict the tasks' transition dynamics and reward functions in potentially problematic ways. Although this is necessary to prove that Boolean algebraic composition results in optimal value functions, in this section we investigate whether these can be practically ignored. In particular, we investigate three restrictions: (i) the requirement that tasks share the same terminal states, (ii) the impact of using dense rewards, and (iii) the requirement that tasks have deterministic transition dynamics.

## 7.1 Four Rooms Experiments

We use the same setup as the experiment outlined in Section 4. We first investigate the difference between using sparse and dense rewards. Our sparse reward function is defined as

$$r_{\text{sparse}}(s, a) = \begin{cases} 2 & \text{if } s \in \mathcal{G} \\ -0.1 & \text{otherwise,} \end{cases}$$

and we use a dense reward function similar to Peng et al. (2019):

$$r_{\text{dense}}(s, a) = \frac{0.1}{|\mathcal{G}|} \sum_{g \in \mathcal{G}} \exp(-\frac{|s - g|^2}{4}) + r_{\text{sparse}}(s, a)$$

Using this dense reward function, we again learn to solve the two base task $M_T$ (reaching the centre of the top two rooms) and $M_L$ (reaching the centre of the left two rooms). We then compose them to solve a variety of tasks, with the resulting value functions illustrated by Figure 2.

| (a) $M_{\text{L}}$ | (b) $M_{\text{T}}$ | (c) $M_{\text{L}} \vee M_{\text{T}}$ | (d) $M_{\text{L}} \wedge M_{\text{T}}$ | (e) $M_{\text{L}} \veebar M_{\text{T}}$ | (f) $M_{\text{L}} \,\overline{\vee}\, M_{\text{T}}$ |

Figure 2: An example of Boolean algebraic composition using the learned extended value functions with dense rewards. The top row shows the extended value functions while the bottom one shows the recovered regular value functions obtained my maximising over goals. Arrows represent the optimal action in a given state. (a–b) The learned optimal goal oriented value functions for the base tasks with dense rewards. (c) Disjunctive composition. (d) Conjunctive composition. (e) Combining operators to model exclusive-or composition. (f) Composition that produces logical nor. We note that the resulting value functions are very similar to those produced in the sparse reward setting.

We also modify the domain so that tasks need not share the same terminating states (that is, if the agent enters a terminating state for a *different* task, the episode does not terminate and the agent can continue as if it were a normal state). This results in four versions of the experiment:

   (i) `sparse reward, same absorbing set`

  (ii) `sparse reward, different absorbing set`

 (iii) `dense reward, same absorbing set`

 (iv) `dense reward, different absorbing set`

We learn extended value functions for each of the above setups, and then compose them to solve each of the $2^4$ tasks representable in the Boolean algebra. We measure each composed value function by

evaluating its policy in the sparse reward setting, averaging results over 100000 episodes. The results are given by Figure 3.

Figure 3: Box plots indicating returns for each of the 16 compositional tasks, and for each of the four variations of the domain. Results are collected over 100000 episodes with random start positions.

Our results indicate that extended value functions learned in the theoretically optimal manner (`sparse reward, same absorbing set`) are indeed optimal. However, for the majority of the tasks, relaxing the restrictions on terminal states and reward functions results in policies that are either identical or very close to optimal.

(a) $sp = 0.1$

(b) $sp = 0.3$

Figure 4: Box plots indicating returns for each of the 16 compositional tasks, and for each of the slip probabilities. Results are collected over 100000 episodes with random start positions.

Finally we investigate the effect of stochastic transition dynamics in addition to dense rewards and different absorbing sets. The domain is modified such that for all tasks there is a slip probability ($sp$) when the agent takes actions in any of the cardinal directions. That is with probability $1\text{-}sp$ the agent goes in the direction it chooses and with probability $sp$ it goes in one of the other 3 chosen uniformly

at random. The results are given in Figure 4. Our results show that even when the transition dynamics are stochastic, the learned extended value functions can be composed to produce policies that are identical or very close to optimal.

In summary, we have shown that our compositional approach offers strong empirical performance, even when the theoretical assumptions are violated.

## 7.2 Function Approximation Experiments

In this section we investigate whether we can again loosen some of the restrictive assumptions when tackling high-dimensional environments. In particular, we run the same experiments as those presented in Section 5, but modify the domain so that (i) tasks need not share the same absorbing set, (ii) the `pickup-up` action is removed since the only terminal states are reaching the desired/goal objects (the agent immediately collects an object when reaching it), and (iii) the position of every object is randomised at the start of each episode.

We first learn to solve three base tasks: collecting purple objects (Figure 5) collecting blue objects (Figure 6) and collecting squares (Figure 7). Notice that because the pickup action is removed, the environment terminates upon touching a desired object and the agent can no longer reach any other object. This results in the large dips in values we observe in the learned extended values. These extended values can now be composed to solve new tasks immediately.

Figure 5: Extended value function for collecting purple objects.

Figure 6: Extended value function for collecting blue objects.

Figure 7: Extended value function for collecting squares.

Similarly to Section 5, we demonstrate composition characterised by disjunction, conjunction and exclusive-or, with the resulting value functions and trajectories illustrated by Figure 9. Since the extended value functions learn how to achieve all terminal states in a task and how desirable those terminal states are, we observe that it can still be leveraged for zero-shot composition even when the terminal states differ between tasks. Figure 8 shows the average returns across random placements of the agent and objects.

Figure 8: Average returns over 1000 episodes for the *Purple*, *Blue*, *Square*, *Purple OR Blue*, *Blue AND Square* and *Blue XOR Square* tasks.

In summary, we have shown that our compositional approach offers strong empirical performance, even when the theoretical assumptions are violated. Finally, we expect that, in general, the errors due to these violations will be far outweighed by the errors due to non-linear function approximation.

(a) Extended value function for disjunctive composition.

(b) Extended value function for conjunctive composition.

(c) Extended value function for exclusive-or composition.

(d) Value function for disjunctive composition.

(e) Value function for conjunctive composition.

(f) Value function for exclusive-or composition.

(g) Trajectories for disjunctive composition (collect blue or purple objects).

(h) Trajectories for conjunctive composition (collect blue squares).

(i) Trajectories for exclusive-or composition (collect blue or square objects, but not blue squares).

Figure 9: Results for the video game environment with relaxed assumptions. We generate value functions to solve the disjunction of blue and purple tasks, and the conjunction and exclusive-or of blue and square tasks.

# 8   Selecting Base Tasks

The Four Rooms domain requires the agent to navigate to one of the centres of the rooms in the environment. Figure 10 illustrates the layout of the environment and the goals the agent must reach.

Figure 10: The layout of the Four Rooms domain. The circles indicate goals the agent must reach. We refer to the goals as `top-left`, `top-right`, `bottom-left`, and `bottom-right`.

Since we know the goals upfront, we can select a minimal set of base tasks by assigning each goal a Boolean number, and then using the columns of the table to select the tasks. To illustrate, we assign Boolean numbers to the goals as follows:

| $x_1$ | $x_2$ | Goals |
|---|---|---|
| $r_\varnothing$ | $r_\varnothing$ | `bottom-right` |
| $r_\varnothing$ | $r_\mathcal{U}$ | `bottom-left` |
| $r_\mathcal{U}$ | $r_\varnothing$ | `top-right` |
| $r_\mathcal{U}$ | $r_\mathcal{U}$ | `top-left` |

Table 1: Assigning labels to the individual goals. The two Boolean variables, $x_1$ and $x_2$, represent the goals for the base tasks the agent will train on.

As there are four goals, we can represent each uniquely with just two Boolean variables. Each column in Table 1 represents a base task, where the set of goals for each task are those goals assigned a value $r_\mathcal{U}$. We thus have two base tasks corresponding to $x_1 = \{\texttt{top-right}, \texttt{top-left}\}$ and $x_2 = \{\texttt{bottom-left}, \texttt{top-left}\}$.

# 9   UVFA Architecture and Hyperparameters

In our experiments, we used a UVFA with the following architecture:

1. Three convolutional layers:
   (a) Layer 1 has 6 input channels, 32 output channels, a kernel size of 8 and a stride of 4.
   (b) Layer 2 has 32 input channels, 64 output channels, a kernel size of 4 and a stride of 2.
   (c) Layer 3 has 64 input channels, 64 output channels, a kernel size of 3 and a stride of 1.
2. Two fully-connected linear layers:
   (a) Layer 1 has input size 3136 and output size 512 and uses a ReLU activation function.
   (b) Layer 2 has input size 512 and output size 4 with no activation function.

We used the ADAM optimiser with batch size 32 and a learning rate of $10^{-4}$. We trained every 4 timesteps and update the target Q-network every 1000 steps. Finally, we used $\epsilon$-greedy exploration, annealing $\epsilon$ to 0.01 over 100000 timesteps.