[Reviews · NeurIPS 2020]

Review 1

Summary and Contributions: This paper considers the multi-task goal reaching setting in RL: that is, families of RL problems that share the same environment dynamics in which maximal reward is obtained by reaching a set of goal states, where different sets correspond to different tasks. As these sets form a boolean algebra, the paper then lifts that Boolean algebra to act on reward and value functions corresponding to these goal state sets. This allows for fast inference of a large number of value functions (exponential in the number of goal states) after learning the value functions for a small number of goal state sets (logarithmic in the number of goal states).

Strengths: This paper is well-written and a pleasant read; the assumptions are clearly stated and the mathematical notation, where non-obvious is defined. The results are, to the best of my knowledge, novel, and the theoretical claims appear to be sound. Perhaps one of the most useful insights of the paper is that, in the shortest-path setting given here, the number of value functions that must be learned before being able to infer the value function of any arbitrary is logarithmic in the number of goal states. I think a lot of the RL community wouldn’t be aware of this fact, and it nicely highlights the power of picking goals that take advantage of the structure of a given problem for more efficient learning and generalization. The experiments confirm the theory in the tabular setting, and the function approximation setting also appears promising.

Weaknesses: The biggest weakness of this paper is the restrictiveness of its assumptions: the contribution is less a boolean algebra for tasks in RL as a boolean algebra for goal state sets in shortest-path problems. While these do constitute a non-negligible subset of RL problems, many tasks of interest are not of this form. It’s not clear how the results from the paper how one would extend this boolean task algebra to other problems in RL, or indeed whether such an extension is possible. The assumptions also make the theory a bit less impressive than what the abstract suggests; given the natural construction of a Boolean algebra from a set of goal states, the extension to task reward functions is quite straightforward. I would have liked to see more empirical analysis of the value function approximation setting (rather than the exact tabular setting). Although it is likely not possible to obtain theoretically rigorous results in settings where the assumptions of the paper are violated, an empirical evaluation demonstrating the utility of the proposed method even in such settings would be encouraging. For example, could you evaluate the approach in the sparse reward setting? The notion of ‘task composition’ used in this paper is less powerful than that used more broadly in RL; the composition of boolean expressions described here is effectively equivalent to defining subsets of the absorbing state set.

Correctness: I have gone through the proofs and they appear to be mostly correct. One concern I have relates to the assumptions. How does one guarantees that an optimal policy terminates in an absorbing state? Presumably this requires that the rewards in the non-absorbing states be sufficiently small relative to the absorbing state, but I didn’t see this discussed in the paper. Code is provided for the empirical evaluation, which is consistent with the theory.

Clarity: The paper is for the most part clearly written. However, it would have been helpful to see a concrete definition of ‘task’ earlier in the paper as the term is somewhat overloaded in the literature.

Relation to Prior Work: Prior work is nicely summarized and the distinction between the paper and previous works is made clear.

Reproducibility: Yes

Additional Feedback: Some ideas on how to relax the restrictive assumptions: The relationship to UVFAs is intriguing, and may potentially lead to a means of applying an approximate version of the results of this paper to more complex settings. For example, what happens if one applies the Boolean operators on value functions to UVFAs? While it’s probably possible to construct MDPs in which this won’t work, it seems plausible that for sparse enough reward settings one might obtain good value function approximations. I also wonder if it might be possible to apply these results to the setting of van Niekirk et al., which appears somewhat looser in the nature of the MDP transition dynamics and the reward function. --------------------------------------------------------------- UPDATE POST-FEEDBACK --------------------------------------------------------------- The authors' feedback has clarified my concerns regarding some of the assumptions from the paper. A couple of points remain that I feel weren't fully addressed by the rebuttal. In R* (i): I (and I believe other reviewers) am interested in seeing what happens when the method for composing tasks performs when it is applied to the estimates \hat{Q}(s, a, g, \theta) from a UVFA in settings that diverge to various degrees from the assumptions made in the paper. I think that further evaluations in this setting showing how far one can push the assumptions would strengthen the paper. The request I made that is addressed in R1 (ii) was (implicitly, and I apologize for not making this clearer in my original review) for more general sparse reward settings that violate the assumptions in more interesting ways than the current environment setting does. It would potentially benefit the paper to further include results that either a) explain why the assumptions are necessary and what breaks in different settings or ideally b) quantify the error obtained by the predicted Q-functions as the assumptions are violated to varying degrees. Ultimately, I still think the paper has some nice results and wouldn't be out of place at NeurIPS, but the limited settings where the theory holds and the lack of empirical results exploring the method's behaviour in more interesting settings prevent me from strongly recommending the paper for acceptance.


Review 2

Summary and Contributions: The paper proposes applying Boolean Algebra to value functions in order to compose learned base tasks for solving new tasks. The paper and the proposed approach are overall sound. Yet the assumptions behind the approach (deterministic transitions, shared absorbing states, learning extended value functions from scratch etc.) as well as the examples are limited. These limitations however are stated clearly in the paper.

Strengths: The principal idea of solving tasks by composing value functions using Boolean Algebra is appealing, since it could enable solving tasks that can be formulated as a combination of the base tasks. The limitations of the method aside, the approach is sound and clearly presented with easy-to-follow examples. The paper also features a clear literature review. While there are limitations to the approach, the class of problems that this approach could be applied to is still decently big.

Weaknesses: The proposed approach is quite limited since it is assuming an MDP with deterministic transition probabilities and shared absorbing states between tasks. This is pointed out fairly clearly in the paper but could maybe made more obvious in a few places. The class of problems that the method can still be applied to is still relevant. Hence, pointing this out would not harm the paper. The presented results are great for understanding the approach in an easy-to-follow way. However, given that the examples are small and there are concerns with the applicability of the approach, it would be great to demonstrate the method on more complex examples to ensure readers that the method scales. At this point, while the basic idea is sound, it is hard to estimate if the method will scale to much more complex problems, such as robotic manipulation, as indicated in the paper. I believe adding such an example would greatly strengthen the paper.

Correctness: The paper and associated claims seem correct.

Clarity: The paper is clearly written and the experimental results help with understanding the approach.

Relation to Prior Work: Prior work is sufficiently discussed and the work sufficiently differentiated from prior work.

Reproducibility: Yes

Additional Feedback: Under broader impact, the paper just states "Not applicable.". Given that this is a theoretical contribution, I understand the motivation behind it. But I think one should at least very briefly outline while this is not applicable.


Review 3

Summary and Contributions: This work formalises a Boolean algebra over tasks. In this algebra new tasks can be defined through intersection, union or negation of previous tasks. Previous work covered only intersection and union. Current work adds negation. The authors define a modified value function on which the same operations yield the optimal modified value-function for a new task given the optimal value functions of the previous tasks. This allows for zero-shot learning of new tasks if they can be derived from learned ones.

Strengths: - Rigorously defined theoretical framework. - Good analysis of the overhead needed to learn the extended value function. - The proposed method is robust, and it does not depend on sensible hyperparameters, etc.

Weaknesses: - The extensive set of assumptions limit the applicability (same terminal states for all tasks, same state space, binary rewards in terminal states). - In order to derive the policy the burden is on the engineer to describe the new task using boolean operations between previously learned ones (not a learning approach). - The abstract starts with a reference to lifelong learning, but there’s no relevant benchmark used. The paper does not address the problems specific to lifelong learning. - No discussion on how to benefit when a new task cannot be described as the result of boolean operations applied to the old ones. - The paper does not address a machine learning problem per se.

Correctness: Yes, the claims and the experiments seem to be correct.

Clarity: Yes.

Relation to Prior Work: The authors acknowledge previous work, but they do not quantitatively compare their results with (Van Niekerk, 2019) at least for the tasks which allow this comparison to be made (Van Niekerk does not cover negation, therefore the number of tasks that benefit from zero-shot transfer are fewer).

Reproducibility: Yes

Additional Feedback: I think a strong point to be made here is that through learning elementary tasks and zero-cost applying the compositionality rules an agent might learn faster (more sample-efficient) a hard task.


Review 4

Summary and Contributions: This paper introduces a method for transferring knowledge from a set of base tasks to a new task, where the new task is expressed as a Boolean operation on the base tasks. These tasks are restricted to share the same deterministic transitions and terminal states. The tasks are allowed to differ only in the reward from the shared terminal states. The authors define Boolean operations over tasks and their proposed extended value functions. The authors then demonstrate that after initially learning the optimal extended value functions for the base tasks, a newly composed task can be solved without additional learning. The ability to solve composed tasks via this zero-shot framework is verified in the tabular 4-doors environment and an image-based environment that requires function approximation to learn the extended Q-value function.

Strengths: The ability to reuse previous knowledge to solve new tasks is appealing for multi-task reinforcement learning. In that regard, a technique to solve arbitrary Boolean combinations of base tasks would be of interest to the community and potentially useful to RL practitioners. The work is motivated and formalised clearly. The idea of creating a framework where value functions can be composed to solve tasks expressed in a Boolean algebra is nice. The explanation for why regular action value functions are insufficient for Boolean composition is also useful. The formal Boolean algebra over tasks might provide insights into related work on logically composing RL tasks. The experiments support the theoretical claims that certain newly composed problems can be solved in a zero-shot manner. It is promising that the method exhibits strong performance even when the theoretical assumptions are dropped in the 4-doors experiment. The experimental results on solving composed tasks with function approximation are also encouraging. The paper is well-written and easy to follow.

Weaknesses: The theoretical analysis requires substantial restrictions on tasks. This raises some concerns regarding the applicability of this approach to more general RL problems. Further experiments without some of these restrictions would help solidify the claim that these assumptions do not hinder performance in practice. While the authors do alleviate some of the concerns with the experiments on the 4-doors environment in the appendix, we would be more confident if we could see similar experiments (with stochastic transitions and/or different absorbing states) in the image-based domain as well. It would be useful to see a visualisation of the return for the composed tasks across the random placements of agent and objects in the image-based environment. It is hard to say with certainty that the zero-shot composition is successful without this information. The extended reward is a high penalty (N) when the agent enters a terminal state of a different task. We wonder if introducing this penalty could lead to unintended consequences, especially with function approximation. Could it deter exploration/goal discovery and lead to sub-optimal solutions? This is probably more applicable when all favourable goals under a task are not equally favourable. A related concern is about problems where the shared terminal states assumption is violated. The authors show successful results on the 4-doors environment with different absorbing states in the appendix 7. But does this remain true when the number of goals increase (like the 40 goal environment), or when a task’s goal is at a critical point (like in the corridor bottleneck). One reason for concern relates to the extended reward. If state S was terminal for a different task but does not need to be terminal for the present task, it seems that the agent could be overly discouraged to cross state S due to the large artificial penalty. What are the authors’ views regarding potential limitations of their approach? Overall, this paper presents a well-formulated idea towards solving an important problem. However, the restrictive assumptions made for simplifying the theory might make it hard to extend this approach to more general RL settings. Comments regarding the limitations of the approach and further experiments without some restrictions in the image-based domain could help to better understand the flexibility of the proposed method. The section on related work must be improved.

Correctness: The claims and methods seem correct. However, as mentioned earlier, to assess the success of zero-shot composition in the function approximation experiment, it would be useful to report the return across different placements of agent and objects.

Clarity: The paper is well-written and easy to follow.

Relation to Prior Work: The authors mention some related work, e.g.: "In this work, we focus on concurrent composition, where existing base skills are combined to produce new skills (Todorov, 2009; Saxe et al., 2017; Haarnoja et al., 2018; Van Niekerk et al., 2019; Hunt et al., 2019; Peng et al., 2019). This differs from other forms of composition, such as options (Sutton et al., 1999) and hierarchical RL (Barto & Mahadevan, 2003), where actions and skills are chained in a temporal sequence." However, hierarchical RL dates back 3 decades, e.g.: [a] C. Watkins (1989). Learning from delayed rewards. PhD thesis, King’s College [b] J. Schmidhuber. Learning to generate sub-goals for action sequences. In T. Kohonen, K. Mäkisara, O. Simula, and J. Kangas, editors, Artificial Neural Networks, pages 967-972. Elsevier Science Publishers B.V., North-Holland, 1991. The authors also write: "We modify deep Q-learning (Mnih et al., 2015) to learn extended action-value functions. Our approach differs in that the network takes a goal state as additional input (again specified as an RGB image)." But a goal state as additional input was introduced 3 decades ago: [c] J. Schmidhuber and R. Huber. Learning to generate artificial fovea trajectories for target detection. International Journal of Neural Systems, 2(1 & 2):135-141, 1991. In fact, the additional inputs as goal states in this work were images, too, just like in the submission by the authors. This should be corrected.

Reproducibility: Yes

Additional Feedback: Minor note - Figure 3(b) does not seem to be directly related to the 4-doors experiment and looks more like a statement about the capabilities of the various approaches. Since this fact has already been stated in the text (Line 211 and 218), perhaps this figure is not required. Response to rebuttal: In their rebuttal, the authors further comment on the usefulness of their contributions and their applicability. The descriptive comparison with UVFA is helpful. We thank the authors for providing the average return plot for the image-based problem and clarifying that the experiments in that environment were conducted with different absorbing states. Regarding the discussion on UVFA: perhaps a helpful experiment could show that the Boolean compositions of EVF (with function approximation) solve a task where composing UVFAs leads to behaviour that is not satisfactory. While the work theoretically shows that composing EVFs is the right thing to do under the given assumptions, such an experiment could visibly demonstrate the benefit of the structure that EVF offers in RL settings that require function approximation. We agree with R1 and R2 that further empirical analysis in more complex environments (with function approximation) will make the paper's utility clearer. With the current set of experiments, we know that the idea works in the restricted setting and, when some of those assumptions are dropped, in tabular settings. However, due to the lack of more demanding experiments, we do not know how far this approach can be extended. We also agree with R1 that the restrictive assumptions make the theory less impressive. That said, this work does make a useful contribution in formalising zero-shot Boolean composition of value functions in a setting where the authors clearly state the assumptions. The authors also promise to include missing related work. The approach seems novel and correct. Some (arguably limited) experiments show that these assumptions could be relaxed a bit in practice. Future papers might build on this work, as it might be a step towards the difficult problem of achieving useful compositional behaviour in RL.

[Author Response · NeurIPS 2020]

We thank the reviewers for their constructive feedback and hope to clarify and address their concerns in this response.

**R\*: Responses that are relevant to all reviewers**

(i) **UVFAs may help with more complex settings. What happens if one applies**

**the Boolean operators on them?** Note that our extended value functions (EVFs) are

a subset of general value functions (GVFs) [4] and UVFAs ($Q(s, g, a, \theta)$) are their

function approximators. So you can think of the function approximators of EVFs as a

subset of UVFAs that have a structure useful for zero-shot composition. This structure

is the decoupling of values per goal, and is illustrated here with the EVF of the "collect

blue objects" task. We will add this explanation in the paper.

(ii) **The extensive set of assumptions limit the applicability (same terminal states**

**for all tasks, same state space, binary rewards in terminal states).** Note that this work is mainly theoretical and

follows previous theoretical work [1]. We show that under the same assumptions as [1] (Assump 1), we improve their

result (optimal union and *approximate* intersection) by obtaining optimal union, intersection and negation (Thm 3).

Note that Assump 1 does not require binary rewards in terminal states (also see discussion after Assump 1). Also please

note that when previous works assume goal reaching tasks share the same transition dynamics, they mean formally the

tasks also share the same absorbing states. If we want to adhere strictly to the theory, then in practice, one can have an

action that the agent chooses to achieve goals. For example, in the four-rooms experiments, we have a 5th action for

"stay", such that a goal position only becomes terminal if the agent chooses to stay in it. This represents the intuition

that if an agent is at the goal location of a different task, and chooses to stay in it, then it has clearly chosen the wrong

behaviour for the current task.

(iii) **General comment.** The usefulness of this line of work is that it shows how to compose value functions to

guarantee zero-shot recovery of useful skills. In [1] these value functions are normal value functions ($Q(s, a)$) and

in this work they are general value functions ($Q(s, g, a)$). Since these value functions have large bodies of work on

learning them, this line of work focuses on their composition after learning to obtain combinatorial explosion of skills.

**R1:** (i) **Given the natural construction of a Boolean algebra from a set of goal states, the extension to task**

**reward functions is quite straightforward.** Note that the Boolean algebra just formalises what previous works have

been saying when they say union and intersection over tasks. The more important contribution here is the zero-shot

composition (Thm 3) and the homomorphism between the Boolean algebra of tasks and Q-values (Cor 2).

(ii) **Could you evaluate in the sparse reward setting?** The function approximation experiment was in this setting.

(iii) **How does one guarantee that an optimal policy terminates in an absorbing state?** This is a standard

assumption in infinite horizon/SSP problems, and is a requirement for policy/value iteration [2].

(iv) **Can these results be applied to the setting of [1].** Our zero-shot results (Thm 3) is in this setting (Assump 1).

**R2:** (i) **It is hard to estimate if the method will scale to much more complex problems, such as robotic manip-**

**ulation.** Please see response R\* (i,iii) above. EVFs are a subset of GVFs which can be viewed as learning N value

functions. If you can learn one (e.g. via PPO) then you can learn them all [4]. Methods like hindsight experience replay

[3] demonstrate this can be done efficiently using UVFAs with any suitable RL algorithm (e.g DQN, PPO, DDPG).

(ii) **Under broader impact, I think one should at least give a brief outline.** We will add the following: Our work

is mainly theoretical, but is a step towards creating agents that can solve tasks through human-understandable Boolean

expressions, which could one day be deployed in practical RL systems.

**R3:** (i) **In order to derive the policy the burden is on the engineer to describe the new task using Boolean**

**operations between previously learned ones (not a learning approach).** Note that this is not a weakness, but rather

a desirable feature of getting to the point where we can program RL agents to solve combinatorially more tasks than

they learn, and do it in a human understandable way! This is the main motivation for this line of work.

(ii) **There's no relevant benchmark for lifelong learning used. The paper does not address the problems specific**

**to lifelong learning.** Note that this line of works (see Sec 6) are steps towards lifelong learning by enabling agents to

solve combinatorially more tasks than they learn. We use the same theoretical setting and experiment domain as [1].

(iii) **Authors do not quantitatively compare their results with (Van Niekerk, 2019)** We did. Please see Figure 3(c).

Here we demonstrated the combinatorial benefit of having zero-shot negation and conjunction in addition to disjunction.

**R4:** (i) **Experiments with different absorbing states in the image-based domain.**

Just as in [1], the experiments we did in the image-based domain was indeed with

different absorbing states (See trajectory to blue circle in Figure 4(a,c) and more in

source code). Also, here is the plot showing average returns (over 10k episodes) for the

composed tasks across the random placements of agent and objects. We will add it to

paper. Also thanks for the additional references which we'll incorporate.

*[1] Van Niekerk et al., Composing Value Functions in RL, 2019; [2] Bertsekas, RL and*

*Optimal Control, 2019; [3] Andrychowicz et al., Hindsight Experience Replay, 2017; [4] Sutton et al., Horde, 2011.*


[Meta-Review · NeurIPS 2020]

All reviewers support acceptance for the contributions, namely the development of a boolean task algebra for reinforcement learning, a clear theoretical and empirical analysis, and efficient zero-shot transfer by task composition when the problem structure is amenable. I also recommend acceptance. Please consider revising your paper to address the concerns raised in the reviews and rebuttal, in particular the comments on the restrictive assumptions.